# A Cluster Randomised Controlled Trial of an Intervention to Increase Physical Activity of Preschool-Aged Children Attending Early Childhood Education and Care: Study Protocol for the ‘Everybody Energise’ Trial

**DOI:** 10.3390/ijerph16214275

**Published:** 2019-11-04

**Authors:** Tessa Delaney, Jacklyn K. Jackson, Jannah Jones, Alix Hall, Ashleigh Dives, Taya Wedesweiler, Libby Campbell, Nicole Nathan, Maria Romiti, Stewart G. Trost, Melanie Lum, Yeshe Colliver, Lara Hernandez, Sze Lin Yoong

**Affiliations:** 1Hunter New England Population Health, Wallsend, New South Wales 2287, Australia; Jacklyn.Jackson@health.nsw.gov.au (J.K.J.); Jannah.Jones@health.nsw.gov.au (J.J.); Ashleigh.Dives@health.nsw.gov.au (A.D.); Taya.Wedesweiler@health.nsw.gov.au (T.W.); Libby.Campbell@health.nsw.gov.au (L.C.); Nicole.Nathan@health.nsw.gov.au (N.N.); Maria.Romiti@health.nsw.gov.au (M.R.); Melanie.Lum@health.nsw.gov.au (M.L.); Serene.Yoong@health.nsw.gov.au (S.L.Y.); 2School of Medicine and Public Health, University of Newcastle, Callaghan, New South Wales 2308, Australia; 3Hunter Medical Research Institute, Newcastle, New South Wales 2300, Australia; Alix.Hall@hmri.org.au; 4Priority Research Centre for Health Behaviour, University of Newcastle, Callaghan, New South Wales 2308, Australia; 5Institute of Health and Biomedical Innovation at Queensland Centre for Children’s Health Research, Queensland University of Technology, Kelvin Grove 4059, Australia; s.trost@qut.edu.au; 6Department of Educational Studies, Macquarie University, North Ryde, New South Wales 2109, Australia; yeshe.colliver@mq.edu.au; 7NSW Office of Preventive Health, Liverpool, New South Wales 2170, Australia; Lara.Hernandez@health.nsw.gov.au

**Keywords:** early childhood education and care, physical activity, preschool, RCT, intervention studies, early childhood, sedentary behaviour

## Abstract

The use of ‘Energisers,’ short bouts of moderate-to-vigorous physical activity (MVPA), have been shown to significantly increase children’s physical activity within the school setting but not within Early Childhood Education and Care (ECEC) centres. The aim of this study is to assess the efficacy of an intervention involving the provision of educator-led daily Energisers to increase the time children spend in MVPA while attending ECEC. Fourteen ECEC centres in the Hunter region of New South Wales, Australia, will be randomised to either an intervention or control group. The intervention group will be supported by the research team to implement three brief (5-min) educator-led Energisers each day for children aged three to six years between the hours of 9:00 a.m. to 3.00 p.m. Control ECEC centres will continue to provide ‘normal practice’ to children. The primary trial outcome is child minutes of MVPA whilst in ECEC, assessed objectively via accelerometery over three days. Outcome assessment will occur at baseline and 6 months post-baseline. Linear mixed models under an intention-to-treat framework will be used to compare differences between groups in MVPA at follow-up. This will be the first cluster randomised controlled trial to test the efficacy of Energisers in isolation on increasing the time children spend in MVPA.

## 1. Introduction

Adequate physical activity (PA) during early childhood is a key determinant of child development, wellbeing, physical fitness and long-term health [1,2]. As such, it is recommended young children (aged three to five years) engage in at least 180 min of PA throughout the day and of this, 60 min should be moderate-to vigorous physical activity (MVPA) [3]. Despite this, global data indicates that less than half of young children adhere to PA recommendations, with even less engaging in sufficient MVPA each day [4,5]. Given that child PA patterns can track longitudinally [6], even into adulthood [7], the promotion of adequate PA in early childhood has been recognised as a public health priority to reduce the future burden of chronic diseases [3].

Centre-based early childhood education and care (ECEC) centres, including long day care and preschools, provide access to a significant proportion of the population aged less than five years for prolonged periods each day, with 93% of Australian children aged four to five years attending formal ECEC [8]. Despite this, systematic review evidence suggests that young children are not sufficiently active during their time in ECEC [9]. Additionally, research in ECEC has found that children spend almost half of their day engaged in sedentary behaviour [10]. As such, ECEC services provide a key setting for intervention to increase child activity.

Previous systematic reviews in this setting identified that physical environment enhancement strategies (e.g., playground markings, equipment, space) as well as multicomponent behavioural strategies, can be effective at improving child PA [11]. However, the impact of multicomponent, behavioural PA interventions have been found to be inconsistent. A recent systematic review found that ECEC interventions that provided increased opportunities for children to participate in structured PA, are delivered by experts and under highly controlled conditions, demonstrated a significant effect on child activity [12]. Such interventions, however, have limited ability to be translated into normal practice due to ongoing costs associated with delivery, complexity and potential incompatibility with ECEC resources. Unsurprisingly, ECEC centres report many barriers to the implementation of such PA interventions [12]. Issues related to the implementation of multi-component behavioural interventions that require significant ongoing effort, time, resources and staff training, have been identified as barriers largely contributing to these findings [13]. As such, there remains significant opportunity to investigate the efficacy of simple, structured PA interventions within ECEC centres that can be delivered with high fidelity by centre staff within normal practice.

‘Energisers’ or brief structured PA sessions, led by an educator may represent one such approach to increase the time children spend in MVPA. Energisers have been shown to be effective in improving children’s PA [14] and cognitive function [15] within the school setting. For example, a systematic review of brief (10–20 min) PA interventions in education settings (primary and secondary) identified 15 studies that examined effectiveness of brief PA sessions on child PA outcomes [16]. While the majority of studies (*N* = 12) reported increases in child PA outcomes post intervention, the review did not identify any ECEC based trials that focused on child PA outcomes [16]. However, there is emerging evidence to support the feasibility and potential efficacy of the implementation of daily Energisers in ECEC [17]. An eight week ECEC based intervention that involved the provision of three, five minute Energisers into the daily routine significantly increased child activity while in ECEC compared with those in the control group at follow up (+4417 Acceleration Units/hour; *p* < 0.001) [17]. While promising, there have been no randomised controlled trials to test the efficacy of such an intervention on increasing child MVPA while in ECEC. Therefore, the primary aim of the current trial is to assess the efficacy of an ECEC-based intervention involving the provision of three, five minute educator led activity breaks (called ‘Energisers’) for increasing the time young children spend in MVPA while in ECEC.

## 2. Materials and Methods

The study methods will be reported in accordance with the CONSORT statement for cluster randomised controlled trials [18] and SPIRIT statement for clinical trial protocols [19]. The trial was prospectively registered with the Australian New Zealand Clinical Trials Registry (reference ACTRN12619000042145).

### 2.1. Design and Setting

The study will employ a parallel cluster randomised controlled trial design. Fourteen ECEC centres located in the Hunter region of New South Wales (NSW), Australia will be randomised to either an intervention or a ‘normal practice’ control group. The Hunter region is a geographically and socioeconomically diverse region encompassing major city, inner regional and outer regional areas [20]. The study will take place in ECEC centres, including pre-schools which typically cater for children aged three to five years and operate between the hours of 9:00 a.m. to 3:30 p.m. and long day care centres which typically cater for children aged six weeks to six years and operate between the hours of 7:00 a.m. and 6:00 p.m. [21]. Approval to conduct the study was obtained from Hunter New England Human Research Ethics Committee (reference number 06/07/26/4.04) and the University of Newcastle Human Research Ethics Committee (reference number H-2008-0343).

### 2.2. Participant Eligibility

#### 2.2.1. Early Childhood Education and Care (ECEC) Centres

To be eligible to participate, ECEC centres (defined as preschools and long day care services) will be located in the Hunter region of NSW, Australia and have a daily enrolment of at least 36 children aged between three to six years. ECEC centres already implementing daily Energisers to facilitate children’s PA, together with ECEC centres currently participating in alternate research trials aiming to increase child nutrition and/or PA, will be excluded from participating in the trial. Additionally, occasional care, mobile childcare services, services catering solely for special needs populations and NSW Department of Education and Communities services (approximately 3% of services) will be excluded as the ethical clearance and intervention design are not appropriate for such services.

#### 2.2.2. Parents and Children

For children to be eligible to participate in the trial, they must be aged three to six years at the time of data collection, attend ECEC on at least one day of the designated data collection period and have active parental consent. Children with an intellectual or physical impairment that may impact on their PA capacity or prevent them from complying with data collection protocols will be ineligible to participate.

### 2.3. Recruitment Procedures

#### 2.3.1. Early Childhood Education and Care (ECEC) Centres

A list of all ECEC centres located within the study region will be obtained from the NSW Ministry of Health and screened for eligibility by the research team (approximately 420 ECEC centres) [22]. A convenience sample of 60 ECEC centres will be invited to participate via mail, email and telephone with recruitment continuing until the required sample of centres (*N* = 14) consent to participate. Specifically, study information statements will be mailed to centre managers inviting study participation. Approximately two weeks later, a research assistant will telephone the centre manager to answer any questions, assess eligibility and invite participation. Centre managers will be invited to provide consent for the research team to (i) approach parents to invite study participation; (ii) complete baseline and follow-up data collection at their centre; and (iii) for their own participation in a survey at baseline and follow-up. The research team has previously employed this recruitment strategy in other ECEC-based trials, obtaining consent rates of over 70% [10,14]. Following consent from the centre manager, the research assistant will schedule a three-day baseline data collection visit. To minimise attrition, ECEC centres will be contacted prior to follow-up data collection to thank them for their participation and to schedule a three-day follow-up data collection visit.

#### 2.3.2. Parents and Children

Following consent from the centre manager, a research assistant will deliver recruitment packs (one per parent of each enrolled child aged three to six years) and a data collection box to each consenting ECEC centre. Centres will be asked to distribute these packs to parents via methods considered most effective and appropriate by the centre manager in communicating with parents (e.g., placed in children’s pigeonholes or lockers or handed directly to parents at drop-off or pick-up). The recruitment packs will contain an information statement and consent form. Parents will be asked to return the consent form to the data collection box located at the centre and provided with the direct phone number of a research assistant if they wish to ask any questions regarding participation. Approximately two to three weeks prior to the scheduled data collection period, trained research assistants will attend each ECEC centre at drop off and/or pick up periods to remind parents about the study and request consent for participation in the study. Parents will be invited to provide active consent for—(i) their child to wear an accelerometer while in ECEC during a three day data collection period at baseline and follow-up; (ii) their child to complete assessments of cognitive function at baseline and follow-up (secondary outcome); and (iii) their own participation in a survey at baseline and follow-up. Parents will be able to select which components of the study they are willing to provide consent for. All data for which consent is received will be used in the assessment of study outcomes.

### 2.4. Randomisation, Allocation and Blinding

Following baseline data collection, ECEC centres will be randomly allocated to the intervention or ‘normal practice’ control group in a 1:1 ratio using a computerised random-number generator in Microsoft Excel^®^. Randomisation of ECEC centres will be stratified into four strata by the socioeconomic status of the area where the centre is located and by the centre type (long day care or preschool) based on evidence of an association between these factors and the PA practices of centres [23,24]. Due to the nature of the intervention, centres will be informed of their experimental group allocation after baseline data collection by a member of the research team. Whilst all efforts will be made to keep data collectors blinded to group allocation, due to the provision of some resources to centres (e.g., Energiser cards), data collectors may become aware of group allocation during attendance at the centre for follow-up data collection. Staff responsible for data analysis will be different individuals and blinded to group allocation.

### 2.5. Intervention

Immediately following baseline data collection and randomisation, centres allocated to receive the intervention will be contacted by a member of the research team to deliver the intervention.

The “Everybody Energise” intervention was designed to increase the time children spend in MVPA while in ECEC through the delivery of brief structured PA sessions (‘Energisers’). Specifically, within a six-hour period (9:00 a.m. to 3:00 p.m.) the intervention will involve educators implementing five minute Energisers, three times daily, to children in the pre-school room/s (aged 3–6 years), using a range of Energiser activities adapted from a variety of sources [25,26,27,28]. The intervention will be operational across the entire study period until the end of follow up data collection (6 months post-baseline). Energisers can be described as educator-led group activities that require short (5-min) bursts of MVPA, requiring minimal (if any) equipment and set up time, as well as being adaptable to indoor and outdoor conditions and confined spaces [29]. Energisers incorporate gross motor skills such as running, jumping, skipping or hopping and can be adapted depending on the ability and age of the child. They may include games (e.g., following the leader); music (e.g., musical statues), screen based games (e.g., dance based games freely available from internet resources including GoNoodle and YouTube); drama (e.g., pretend to be an aeroplane taking off) and following instructions (e.g., when educator calls out ‘green light’ children are to run quickly).

To support the implementation of daily Energisers in the Everybody Energise trial, ECEC centres will be provided with a suite of 60 suggested Energiser activities that were assessed by an educator as appropriate for children aged three to six years old. The suggested Energisers were designed to target MVPA and known facilitators to the implementation of Energisers in the school setting. For example, previous research suggests that Energisers need to be (i) easily accessible to educators (via printed instruction cards) and (ii) have the option to be flexibly administered (e.g., allowing educators to choose when they are scheduled, where they are delivered and from a range of Energiser activities that best suit the ability and interest of the children) [14,30]. Furthermore, five minute Energisers, three times daily, of moderate intensity activity, are considered to be acceptable and feasible to teachers [14]. As such, the Everybody Energise intervention will provide centres with a box of 60 “Energiser Activity Cards.” The Energiser cards are designed to be easily adaptable (i.e., can be delivered in small or large spaces, indoor or outdoor, equipment or no-equipment) and modified to be more or less complex to accommodate for the varied ability and age of children. The Energiser cards include simple instructions on how to deliver the Energiser activity and will be sorted into five categories including; games requiring no equipment (e.g., jumping and skipping), equipment-based games using equipment already available in the centre (e.g., newspaper, bubbles, balls), screen based games (e.g., YouTube and GoNoodle), music-based games which can be played to any music chosen by the centre and favourites (i.e., blank cards from which the other four categories can be added for quick access). Examples of the Energiser Activity Cards can be found in Table 1. Educators will be encouraged to use the cards to deliver three Energiser sessions each day with each session being approximately five minutes in duration. The scheduling of the Energisers will be at the discretion of the centre as will the selection of Energiser activities. Therefore, the type of Energiser and time and sequence with which they are delivered throughout the ECEC day (9:00 a.m. to 3:00 p.m.) may vary between centres. However, all activities are designed to target MVPA.

To improve adherence to the intervention, centres will also be provided with the following support by the research team. The support was specifically designed to target known barriers to delivering PA interventions in ECEC centres using the Behaviour Change Wheel [31] and will involve:Obtaining centre manager executive support of the intervention;Face-to-face educational meeting with the centre manager and lead educators;Local consensus processes with the centre manager and educators;Provision of educational resources designed specifically for this study; andTailored telephone support from the research team on a minimum of two occasions throughout the intervention period.

Further details regarding support strategies can be found in Table 2.

### 2.6. Control

ECEC centres allocated to the control group will not have access to any of the intervention materials during the intervention period. ECEC centres in the control group will continue to receive ‘normal practice’ as part of a state-wide obesity prevention program [32] during the intervention period. This may include the research team contacting the ECEC centre to assess service implementation of nutrition and PA practices and the provision of nutrition-related support and resources on a per request basis. Data regarding centres’ exposure to such support and other potential sources of contamination will be assessed at follow-up. At the conclusion of follow-up data collection, the control centres will be provided with the intervention material (the ‘Energiser resources’).

### 2.7. Data Collection and Measures

#### 2.7.1. Primary Trial Outcome

The primary trial outcome is the mean minutes/day that children spend in MVPA during ECEC. Minutes of MVPA will be objectively assessed via an ActiGraph GT3X+ accelerometer (ActiGraph Corporation, Pensacola, FL) using cut points developed by Pate and colleagues [33]. Accelerometery has been shown to be a valid and reliable method for measurement of PA and is the preferred method for the assessment of PA in children aged three to six years [34].

Consistent with other ECEC trials conducted by the research team [35,36], a day will be considered a “valid monitoring day” if daily wear time is at least 50% (180 min) of the day in ECEC (9:00 a.m. to 3:00 p.m.). Non-wear time will be defined as intervals with at least 20 consecutive minutes of zero counts. Wear time will be calculated by subtracting non-wear time from the total monitoring time for the day. The accelerometers will be worn by children from the time they arrive at the ECEC centre until 3:00 p.m. (or earlier if the child departed the centre) on each day of attendance. A standardised monitoring end time of 3:00 p.m. was chosen to align with typical operating hours of the pre-school centres included in the study. Trained data collectors (blinded to group allocation) will attend centres during the data collection period to fit and collect the accelerometers using a standardised protocol. Accelerometers will be placed above the right iliac crest at the hip using an elastic waistband. Accelerometer data will be collected on three consecutive days during one week at baseline and follow-up.

#### 2.7.2. Secondary Trial Outcomes

##### Other Child Physical Activity Outcomes

The following PA outcomes will also be assessed via accelerometer at baseline and 6 months post-baseline, including:Total daily PA while in ECEC, calculated as the sum of light, moderate and vigorous activity.Total daily activity counts while in ECEC.Mean daily minutes in sedentary behaviour, using recommended cut points by Pate and colleagues [33].

##### Child Cognitive Function

Child cognitive function will be assessed using five tests from the validated Early Years Toolbox app that utilises brief iPad based games to assess (i) inhibition; (ii) visual-spatial working memory; (iii) cognitive shifting; (iv) phonological working memory; and (v) executive functioning and vocabulary [37]. The Early Years Toolbox was developed specifically for young children and utilises game-like assessments, imagery and developmentally appropriate visual and audio instructions to measure child cognitive function. Specifically, ‘Go/No Go’ will be used to assess the ability to inhibit or control behavioural urges. ‘Mr Ant’ will be used to assess visual spatial working memory or the amount of visual information that can be simultaneously retained in the mind; ‘Card Sorting’ will be used to assess cognitive shifting, including the ability to control and re-direct attention; ‘Not This’ will be used to assess phonological working memory or the amount of auditory information that can be concurrently processed in the mind; and ‘Expressive Vocabulary’ will be used to assess executive function and vocabulary, including the ability to identify and name objects. Data collectors will administer the five tests to children via iPad over the data collection period at baseline and follow-up. Tests will be conducted in a quiet and private location within the ECEC centre.

### 2.8. Characteristics of Sample

#### 2.8.1. Centre Characteristics

At baseline, a survey will be conducted with centre managers at participating ECEC centres to assess centre characteristics including—type of centre (preschool or long day care centre); postcode; days and hours of operation; the total number of three to six year-old children enrolled; and the total number of primary contact teaching staff, as well as information on training and education opportunities provided to ECEC staff, centre communication with families relating to PA and centre policy and programming. The survey will be sent to centre managers electronically prior to the data collection period, with any incomplete surveys to be administered by data collectors on one of the days of data collection. Alternatively, centre managers can return this survey via email within two weeks post-data collection. The items used to assess centre characteristics have been used in previous surveys of ECEC centres conducted by the research team [35,36].

#### 2.8.2. Child and Parent Characteristics

At recruitment, parent and child demographic information will be collected via the participant consent form. Specifically, the consent form will collect information regarding child age, sex, child days of attendance at the participating ECEC centre, postcode of residence, parent contact details and nature of parent-child relationship. Additional parent characteristic information regarding age, gender, educational level, household income level, living arrangements, language/s spoken at home and whether the parent identifies as Aboriginal or Torres Strait Islander will be collected via a survey administered online, via pen and paper or telephone interview. Where families provide consent for more than one child aged three to six years to participate in the study, they will be asked to complete the survey based on the eldest consenting child. If a parent completes a survey for more than one child, the data for the eldest eligible child will be used. In the event of a parent completing a survey for each twin or triplet, a randomly selected survey will be used.

### 2.9. Additional Measures

#### 2.9.1. Usual Child Physical Activity

To assess any compensatory changes in PA which could occur outside the hours of ECEC as a result of the intervention, the parent surveys at baseline and follow-up will include items to assess child PA, drawn from the validated Preschool-age Children’s Physical Activity Questionnaire (Pre-PAQ) [38].

#### 2.9.2. ECEC Centre Physical Activity Policies and Practices

At baseline and follow-up, direct observations will be conducted by trained data collectors on one randomly selected day during the designated data collection period at each participating ECEC centre. Data collectors will collect information regarding the ECEC centre PA policies, practices and environment using a modified version of the validated Environment and Policy Assessment and Observation (EPAO) tool [39], which is considered the gold standard for environmental observations conducted in the ECEC setting. The items observed will include active play opportunities, provision of educator-led structured PA and fundamental movement skills sessions, sedentary opportunities and environment, availability of fixed and portable play equipment, staff behaviours and practices (e.g., role modelling and provision of prompts and positive statements to increase PA) and existence of a written PA policy. Information will also be collected regarding (i) minimum and maximum daily temperatures collected via local meteorological bureau website [40]; and (ii) daily UV index collected via the Australian Radiation Protection and Nuclear Safety Agency website [41]. Such factors have been associated with child PA in ECEC and have been included to provide contextual information to enable assessment of broader generalisability and to aid the interpretation of trial findings [42].

#### 2.9.3. Fidelity of Delivery of Energisers

An assessment of the degree to which Energisers were implemented as intended at ECEC centres allocated to the intervention group will be undertaken. Specifically, centre delivery of three daily Energisers throughout the intervention will be assessed via project records maintained by intervention centres and a survey with the lead educator and centre manager at follow up. Furthermore, an assessment of the implementation of Energisers, including their frequency and duration, will be undertaken in both intervention and control centres during one randomly selected day of data collection at follow up using the EPAO tool. A member of the research team will also informally check the fidelity of Energiser implementation by providing at least two phone calls to intervention ECEC centres and provide additional support to those experiencing barriers to implementing three Energisers per ECEC day.

#### 2.9.4. Acceptability of Energisers and Intervention Support to Centre Managers and Lead Educators

Centre managers and lead educators in the intervention group will be invited to complete a survey at follow-up to assess the acceptability of the ‘Energisers’ and support provided by the research team. Participants will be asked to respond on a five-point Likert scale (strongly agree; agree; neither agree nor disagree; disagree; strongly disagree) to a series of statements assessing the acceptability of the ‘Energisers’ and intervention support provided to centres using validated measures of acceptability [43] and items previously utilised by the research team [44]. The survey will be sent to centre managers and lead educators electronically prior to the follow-up data collection period, with any incomplete surveys to be administered by data collectors on one of the days of follow-up data collection.

#### 2.9.5. Fidelity of Intervention Support by the Research Team

The delivery of other intervention components (e.g., educational meeting) as intended, will be assessed using internal project records maintained by the research team.

#### 2.9.6. Adverse Effects

To identify any potential unintended adverse effects of the Energisers, the number of child injuries requiring documentation during the past six months will be assessed during a survey of centre managers at baseline and at follow-up. The items used to assess child injury rates have been previously utilised by the research team in ECEC PA research trials [35,36].

### 2.10. Analysis

An intention to treat approach to analysis will be undertaken. A linear mixed model will be used to assess differences in child MVPA across the ECEC day between the intervention group and control groups at follow-up. The models will include fixed effects for baseline MVPA, group (intervention or control group) and child level characteristics such as child age and sex. A random level intercept for centre will also be included. Multiple imputation will be used to impute missing values. Sensitivity analyses will be undertaken to compare results obtained from analyses of the imputed data to those obtained from complete case analyses. Exploratory subgroup analyses will also be performed for the primary trial outcome for age and sex of the child.

### 2.11. Sample Size and Power Calculation

The study will approach approximately 420 children from 14 ECEC centres across the study region. Assuming the standard deviation of MVPA is 21 min and assuming an intraclass correlation coefficient of 0.07, a sample of 30 children per cluster (*N* = 420 children) will be sufficient to detect a change of 10 min in MVPA in ECEC, with 80% power and an alpha of 0.05. An increase of 10 min of MVPA in children aged three to six years old has been found to have clinically significant, beneficial effects on body fat and peak bone mass [45,46].

## 3. Discussion

This is the first randomised trial to assess the efficacy of an ECEC-based intervention involving the provision of daily educator-led Energisers (specifically three, five-minute Energisers each day) for increasing the time children spend in MVPA while in ECEC. This trial protocol describes the “Everybody Energise” trial with regards to rationale, study design, sample eligibility criteria, recruitment, intervention content and design, outcome measurements and planned analyses.

Brief, structured PA sessions were chosen for investigation as they may represent a simple and feasible strategy for improving child MVPA during the ECEC day. For example, previous research [14,30] suggests that ECEC interventions acceptable to ECEC educators need to be easily accessible and have the ability to be flexibly administered. As such, the Energisers were made to be easily accessible to ECEC educators via the provision of the Energiser activity cards, Energisers were also designed to require minimal set-up time. The Energisers can be flexible in their administration as they require minimal (if any) equipment, can be adapted to indoor and outdoor conditions, can be adapted to the space available within the ECEC centre and are adaptable for the ability of the children. The research team for the “Everybody Energise” trial engaged ECEC educators (the end-user) when developing the Energiser activity cards with the aim of improving intervention feasibility and relevance. Further to support the efficacy of Energiser delivery in ECEC centres, a variety of implementation support strategies were provided to intervention ECEC centres by the research team.

The study findings resulting from this trial will need to be considered in the context of the trial methods. The strengths of this study include the experimental design, the use of validated outcomes measures and central random assignment to study groups. This study is also powered to detect small but meaningful changes to child MVPA. However, there are a number of limitations of this study. First, we adopted a convenience sampling approach in one region of NSW, therefore limiting external validity of findings, however we have sought to include different ECEC types across socioeconomic economic status strata, which is a strength. Second, our procedures for data collection within ECEC centres at baseline and follow-up may lead to incomplete data collection given factors including children not attending on data collection days or newly eligible children to centres contributing follow-up data only. This limitation will be adjusted for by applying imputations of missing values in the intention to treat analysis.

## 4. Conclusions

Notwithstanding these limitations, it is anticipated the findings of this study will significantly add to the current evidence base and will be used by policy makers and practitioners to inform the future implementation of PA interventions within the ECEC setting. If shown to be effective, these findings would support the need for additional efforts to implement the Everybody Energise intervention within ECEC centres at scale.

## Figures and Tables

**Table 1 ijerph-16-04275-t001:** Example of Energiser Activity Cards.

Energiser Category	Energiser Title	Instruction
No Equipment	Thapumpan ‘tha-pum-pan’ (A Traditional Indigenous Game)	Have all the children spread out in the activity area Choose one child to be the “thapumpan” (shark) and have them hold one hand on top of the other like a thapumpan fin The thapumpan then tries to tag the other children while moving like a thapumpan in water When the thapumpan tags another child, that child is the new thapumpan and the game continues
Equipment	Chase and Rescue	Have all the children spread out in the activity area Choose one child as the “Chaser” and two children as the “Rescuers” and give each Rescuer a soft ball All of the children run around When caught by the Chaser the child stands still with their feet apart. They can return to the game when one of the Rescuers has rolled the ball between their legs
Music	Copy Cat	Have all children stand in a circle with one child in the middle Play music and have the middle child choose a dance move that everyone in the circle has to copy Swapping around so that everyone gets a turn in the middle
Screen	Blazer Fresh 1	Have all the children spread out in the activity area Search “Banana Banana Meatball” on GoNoodle and children to follow video instruction Then search “Dynamite” on GoNoodle and children to follow video instruction

**Table 2 ijerph-16-04275-t002:** Support delivered as part of the Everybody Energise trial.

Intervention Component	Description of Support/Intervention Component	Resources and Delivery Mode
Executive Support	The centre manager will be asked to demonstrate executive support to staff within their centre by inviting lead educators (e.g., educational leader and/or room leaders) to a meeting with a member of the research team. The meeting will provide implementation support by providing input on how the program can be implemented at their centre. The centre manager will also be asked to communicate their endorsement to educators and families in written and verbal formats, via their usual communication channels (e.g., email, parent newsletter snippets and within staff meetings).	Telephone and face-to-face support provided to centre managers. Hard copy and electronic support resources for families and staff.
Educational Meeting	A member of the research team will conduct a face-to-face educational meeting involving lead educators within the centre to: Identify potential Energiser scheduling opportunities within the centres’ daily routine. Build knowledge about the importance and benefits of physical activity and low staff training requirements for delivering energisers. Draft centre specific actions plans. This includes documenting agreed goals and actions required to achieve full-implementation of the program. Nominate a program champion to drive centre implementation of the program.	Face-to-face meeting with the lead educators conducted by a member of research team. Centre specific action plan drafted by lead Educators.
Local Consensus Process	Centre managers will allocate one usual staff meeting to facilitate local consensus processes among all centre staff. Consistent with best practice principles, this meeting will commence with a presentation delivered by a member of the research team, followed by group discussion with staff. The aim of this meeting is to: Increase educator knowledge and motivation to deliver the program. Facilitate demonstration of Energisers, highlighting the efficiency to set up and running of Energisers, as well as their flexibility (e.g., they can be performed outside, inside, small spaces, equipment or no equipment). Encourage group agreement to implement the program, with a focus on ‘how’ the program could be implemented within their centre. Facilitate discussion around problem solving barriers to program implementation and update the centre action plan.	Face to face meeting, incorporating support material in the form of a PowerPoint Presentation, Energiser Resources and Action Plan.
Provision of Educational Resources	In addition to the “Energiser Activity Cards,” a suite of additional printed and electronic information and educational resources to support the Energisers will be provided to each centre electronically and as a hard copy. These resources will include fact sheets; programming suggestions; a form to record daily Energiser completion; newsletter snippets; and posters.	Additional resources: posters, newsletter snippets, challenges and solutions document, GoNoodle instructions, fact sheets, daily program examples, Energiser Record form
Tailored Telephone Support	A member of the research team will provide ongoing support to the centre champion and centre manager to assess intervention fidelity and provide the opportunity to give feedback to those centres experiencing any barriers to implementation. These contacts will be via telephone, email or face-to-face visit as required by the centre, with a minimum of two contacts scheduled during the intervention period. Each contact will draw on continuous quality improvement principles to review progress according to a previously developed action plan, provide positive reinforcement and facilitate reflection. Support from the research team may also involve problem solving any new barriers and provide practical advice and additional resources when requested. Additional support from the research team will be provided as needed.	Telephone support call outline/checklist

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
