# Peer review of "A Cluster Randomised Controlled Trial of an Intervention to Increase Physical Activity of Preschool-Aged Children Attending Early Childhood Education and Care: Study Protocol for the ‘Everybody Energise’ Trial"

_ijerph, 2019, doi:10.3390/ijerph16214275_

Round 1

Reviewer 1 Report

The protocol entitled “A cluster randomised controlled trial of an intervention to increase physical activity of pre-school aged children attending early childhood education and care: Study protocol for the ‘Everybody Energise’ trial” proposes a exercising program for children of 3-6 years old in child care centers of a certain region of Australia. According to the authors, the “energisers” will provide a new way-of-life to the children for introducing them in early-stage exercises activities. The proposal is very interesting. Some points have to be clarified:

What is the difference between “educator” and “teacher” in the protocol? According to the document, three activities have to be applied during the day. Is there a classification of the activities taking account of the metabolic rate? If not, it should be. Otherwise, how can differentiate the 3-6 years old activities with activities for older children? What about the activities the children do in their own homes? It is somehow considered? Please discus this subject.

Author Response

Reviewer 1 Comments

Author Response

The protocol entitled “A cluster randomised controlled trial of an intervention to increase physical activity of pre-school aged children attending early childhood education and care: Study protocol for the ‘Everybody Energise’ trial” proposes a exercising program for children of 3-6 years old in child care centers of a certain region of Australia.

According to the authors, the “energisers” will provide a new way-of-life to the children for introducing them in early-stage exercises activities. The proposal is very interesting. Some points have to be clarified:

The authors would like to thank the reviewer for their insightful comments, and are pleased to find the reviewer finds this trial protocol to be of interest.

What is the difference between “educator” and “teacher” in the protocol?

Thank you for identifying this inconsistency. In the previous manuscript the term “educator” and “teacher” was used interchangeably.

Please find in the revised manuscript, these terms have been formatted to be consistently referred to as “educator” throughout.

This is highlighted in red : line 192 and 193

According to the document, three activities have to be applied during the day. Is there a classification of the activities taking account of the metabolic rate? If not, it should be. Otherwise, how can differentiate the 3-6 years old activities with activities for older children? What about the activities the children do in their own homes? It is somehow considered? Please discus this subject.

The authors would like to thank the reviewer for raising an interesting point.

We would like to point out that the accelerometer cut-points used for classifying each 15 second time period as sedentary, light, moderate or vigorous intensity physical activity were established specifically for pre-schoolers (aged 3-5 years) by calibrating accelerometer counts to directly measured energy expenditure. The cut-off points used for this trial are therefore taking into account the higher resting metabolic rate and poorer economy of young children. The cut-off points have been subsequently validated in independent samples of preschool-aged children. The validation paper by Janssen et al (2013) can be found at: doi:10.1371/journal.pone.0079124.

In addition, we will control for the age of children and do a subgroup analysis by child age to see if the intervention is more effective in older vs younger aged children.

As for the activities children do in their own homes, we will capture this data using the Preschool-age Children’s Physical Activity Questionnaire (Pre-PAQ). Details for this have been provided in section 2.9.1. (line 306-308).

Reviewer 2 Report

I was honored to review the manuscript entitled “A cluster randomised controlled trial of an intervention to increase physical activity of preschool-aged children attending early childhood education and care: Study protocol for the ‘Everybody Energise’ trial” submitted to International Journal of Environmental Research and Public Health.

I recommend to accept the manuscript after minor revision.

There are also some other points to correct:

 - please provide the list of abbreviations.

 - Introduction and Discussion section needs improvement- please cite: doi: 10.1097/MD.0000000000014909. ; Biomed Environ Sci. 2016 Oct;29(10):706-712. ; 10.1016/j.biopha.2016.02.017; especially discussion is too short

 - where are conclusions?

 - In discussion please provide “study strong points” and “study limitation” section.

I recommend to accept the manuscript after minor revision.

Author Response

Reviewer 2 Comments

Author Response

I was honored to review the manuscript entitled “A cluster randomised controlled trial of an intervention to increase physical activity of preschool-aged children attending early childhood education and care: Study protocol for the ‘Everybody Energise’ trial” submitted to International Journal of Environmental Research and Public Health.

I recommend to accept the manuscript after minor revision.

The author would like to thank the reviewer for their encouraging comments.

Please provide the list of abbreviations.

To the authors knowledge there is no place for a list of abbreviations within the submitted manuscript (as per journal formatting guidelines).

However, the authors are happy to provide a list of abbreviations here:

ECEC: Early Childhood Educator and Care

EPAO: Environmental and Policy Assessment and Observation

MVPA: Moderate-to-Vigorous Physical Activity

NSW: New South Wales

PA: Physical Activity

RCT: Randomised Controlled Trial

Introduction and Discussion section needs improvement- please cite: doi: 10.1097/MD.0000000000014909. ; Biomed Environ Sci. 2016 Oct;29(10):706-712. ; 10.1016/j.biopha.2016.02.017; especially discussion is too short

Thank you for this list of suggested references.

The authors note that these suggested citations are related to endurance-strength training and mineral status/renal function in obese women.

Given that physical activity literature in children is relatively well developed, and given that the suggested references do not hold direct relevance to our research topic, the authors have opted to not cite these references within the manuscript.

We have however made improvements to the discussion. See page 7, line 369-397.

Where are conclusions?

Thank you for this comment.

Please find we have added a conclusion to this protocol manuscript.

Line: 399-403.

In discussion please provide “study strong points” and “study limitation” section.

Thank you for the suggestion to develop the discussion in this manuscript.

Please find that the discussion has been built-on, and has highlighted “study strong points” (e.g. experimental study design, valid outcome measures, and central random assignment of study groups) and “study limitations” (e.g. convenience sample).

This can be found line: 388-397

Reviewer 3 Report

Actually the manuscript presented is an outline of an intervention project to be carried out. But the project is not carried out, therefore I do not see the interest of publishing a project that has not been carried out. I recommend the authors of the project to carry out the intervention and try to publish the results obtained, because the idea is good, but it must be carried out, since if it is not done it makes no sense. I encourage the authors to carry out the intervention and publish it, but I sincerely believe that the submitted manuscript lacks scientific interest.

Author Response

Reviewer Comment 3

Author Response

Actually the manuscript presented is an outline of an intervention project to be carried out. But the project is not carried out, therefore I do not see the interest of publishing a project that has not been carried out. I recommend the authors of the project to carry out the intervention and try to publish the results obtained, because the idea is good, but it must be carried out, since if it is not done it makes no sense. I encourage the authors to carry out the intervention and publish it, but I sincerely believe that the submitted manuscript lacks scientific interest.

The authors acknowledge the reviewers concerns, and will indeed publish their study results after the completion of the trial.

The lead author for this manuscript sought clarification from the special issue editors as to whether a protocol paper would be considered for publication and were encouraged to submit the manuscript for review for the special issue: Interventions to Promote Healthy Movement Behaviours in Early Childhood Education and Care Settings.

The authors would also like to note that the publishing of clinical trial protocols are important for ensuring study conduct, review and reporting, they can also help to improve the reporting of study findings, as well as prevent risk of study duplication.

The authors also note that this journal has previously published study protocols (e.g. https://doi.org/10.3390/ijerph16193548).

Round 2

Reviewer 3 Report

I reiterate my opinion. The study only expresses the idea of ​​creating an intervention program. But it is not carried out, therefore it lacks scientific interest.